# Formation Mechanism of Micro- and Nanocrystalline Surface Layers in Titanium and Aluminum Alloys in Electron Beam Irradiation

**Sergei Nevskii [1,2], Vladimir Sarychev [2], Sergey Konovalov [1,3,\*], Alexey Granovskii [2] and Victor Gromov [2]**

1 Department of Mechanical Engineering, Nanjing University of Science and Technology, 200 Xiao Ling Wei, Nanjing 210094, China; nevskiy_sa@physics.sibsiu.ru

2 Department of Natural Science, Siberian State Industrial University, 42, Kirov Street, 654007 Novokuznetsk, Russia; sarychev_vd@mail.ru (V.S.); legatokun@gmail.com (A.G.); gromov@physics.sibsiu.ru (V.G.)

3 Department of Materials Technology and Aviation Materials Science, Samara National Research University, 34, Moskovskoye shosse, 443086 Samara, Russia

\* Correspondence: ksv@ssau.ru; Tel.: +7-903-943-9999

**Abstract:** The reported study discusses the formation of micro- and nanocrystalline surface layers in alloys on the example of Ti-Y and Al-Si-Y systems irradiated by electron beams. The study has established a crystallization mechanism of molten layers in the micro-and nanodimensional range, which involves a variety of hydrodynamic instabilities developing on the plasma–melt interface. As suggested, micro- and nanostructures form due to the combination of thermocapillary, concentration and capillary, evaporation and capillary and thermoelectric instabilities. This mechanism has provided the foundation for a mathematical model to describe the development of structures in focus in the electron beam irradiation. The study has pointed out that thermoelectric field strength $E \geq 10^6$ V/m is attributed to the occurring combination of instabilities in micro- and nanodimensional ranges. A full dispersion equation of perturbations on the melt surface was analyzed.

**Keywords:** electron beam irradiation; yttrium doped; thermoelectric phenomena; thermocapillary instability; concentration-capillary instability; evaporation pressure

## 1. Introduction

The development of pulsed high intensity technologies with the purpose of improving strength and tribology characteristics of high-duty machine elements represents one of the challenging issues confronting the present-day material science [1]. The material fracture is known to arise in the surface and near-surface layers because long-ranging stresses in them exceed the material strength limit [2]. It is necessary therefore to generate such a structure in the material surface, which would ensure high strength properties. Micro-, submicro- and nanocrystalline structures with a high level of hardness and wear resistance meet these requirements [3]. To form these layers concentrated energy flows are applied i.e., the heterogeneous plasma flow processing in conjunction with the succeeding low energy high current electron beam irradiation [4–6]. The latest research in this field [7,8] has highlighted micro- and nanocrystalline layers with a columnar structure (Figure 1) form when irradiating titanium and silumin processed prior by the electric explosion plasma of yttrium powder. A thickness of a columnar crystallization layer is within a range from 1 to 1.5 μm (Figure 1a) in the Ti-Y system. Columns are inclined to the irradiation direction. This inclination may be connected with the longitudinal melt velocity, which, in turn, causes the Kelvin–Helmholtz instability. Transverse dimensions of columns vary from 100 to 300 nm (Figure 1b) and depend on the irradiation conditions. Above this layer there

is an amorphous layer. The fact that amorphous layers are there in yttrium-doped titanium may be linked to high heating and cooling velocities in the electron beam irradiation $\approx 10^6$–$10^{10}$ K/s. This, in turn, c auses the heating and cooling curves and start/end curves of crystal phase transition to not intersect on the isothermal diagram of phase transition, and enter the amorphous region as soon as a glass transition temperature is reached [9].

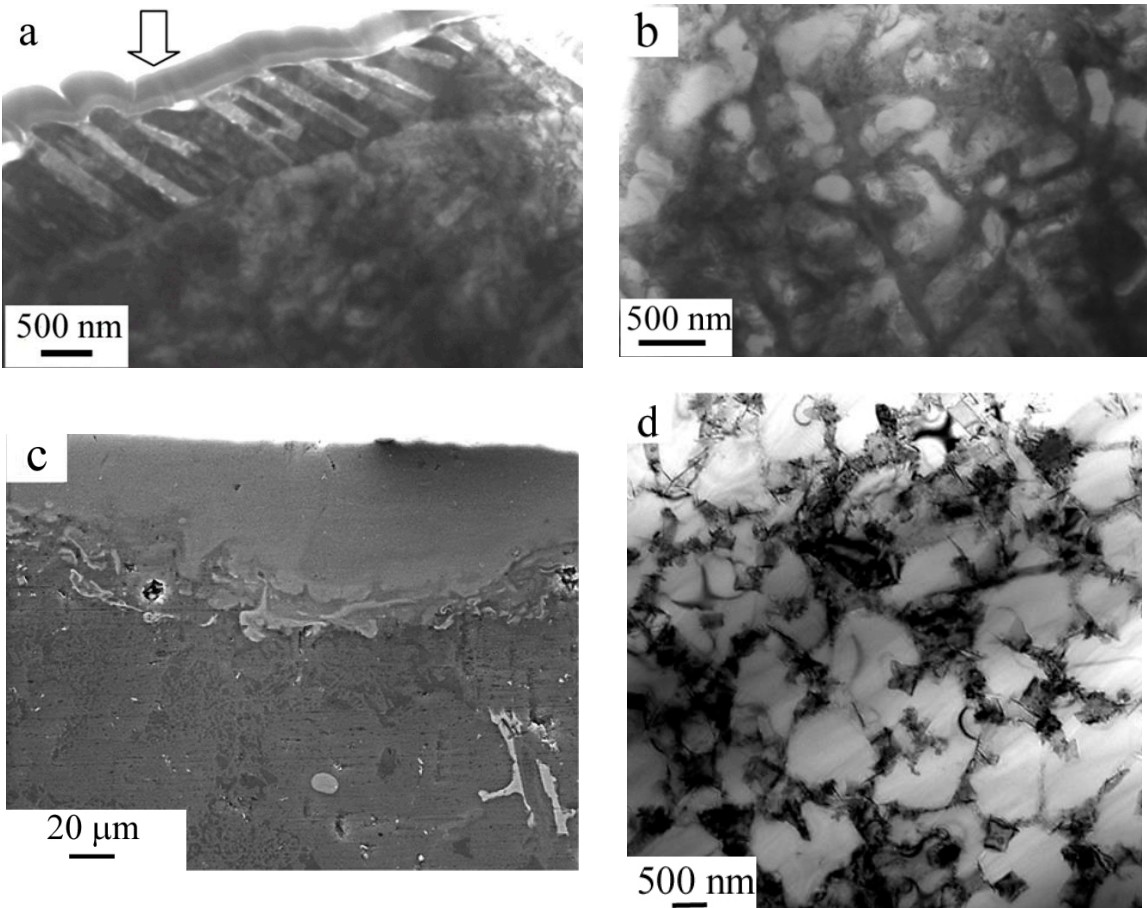

**Figure 1.** The structure of yttrium-doped titanium (**a**,**b**), and silumin (**c**,**d**) surfaces irradiated by an electron beam.

A layer with cell crystallization in the Al-Si-Y system was determined to be 80 μm thick (Figure 1c) [8]. Cell dimensions vary from 0.8 μm to 1.3 μm. (Figure 1d) [8]. An amorphous structure was detected in several zones of the surface layer, which is demonstrated by the halo effect on electron diffraction micro-patterns [8]. For silumin, cooling curves and isothermal curves of phase transition intersect; as a result, the volume of part of the crystal phase grows faster according to the Kolmogorov–Avrami formulation than the amorphous one. This approach including the idea that new phase nuclei arise sporadically owing to thermal fluctuations, makes it possible to follow the crystallization kinetics and estimate critical nuclei dimensions. However, it leaves open the questions of why a structure with unimodal and bimodal grain distribution (subgrains, second phase particles, etc.) forms and doping elements spread over grain boundaries and junctions. In our opinion, these issues can be resolved by the hydrodynamic approach [10–12]. It claims a crystal structure forms affected by a variety of hydrodynamic instabilities on the plasma–melt and melt–crystal interfaces. Small perturbations tend to occur on these boundaries because of thermal fluctuations [13,14]. To begin with, we assume these perturbations are harmonic. These perturbations increase under the instability initiating conditions, i.e., a positive real part of the complex frequency, referred to as a growth rate

$\alpha$. A wavelength, when $\alpha = 0$, is called a critical $\lambda_{cr}$ and instability starts developing from this point. It influences an initial size of structure elements. If a growth rate is maximal for a $\lambda$, this wavelength governs the most probable size of structure elements. It is a well-known fact that the gradient between temperature and concentration of doping elements arises in a molten material when processed by concentrated energy flows [15,16], resulting in thermal concentration and capillary instability. This instability is supposed to be one of the reasons for short wave micro- and nanometer periodic structures. If a material is irradiated in high vacuum conditions, the vapor return pressure is of significant importance, causing evaporation and capillary instability. The study [17] says a contribution of the evaporation pressure is decisive to the instability of capillary waves if a constraint $k_p \geq -\frac{p'_v}{\sigma_T}$ is true, where $p'_v$ and $\sigma_T$—thermal coefficients of evaporation pressure and surface tension, respectively.

Moreover, thermal and concentration and capillary instabilities thermoelectric effects caused by a forming electric field are important to high temperature gradients ~$10^6$–$10^{12}$ K/m [18–20]. Once a temperature fluctuation arises, fluctuations of the electric field and its charge are registered. The effect of a constant electric field on this charge is critical for the power and instability provided so that the constant temperature gradient is high. The movement amplitude depends on the physical parameters of a liquid in focus and an active temperature gradient. The data on a molten layer thickness, when the thermal electric convection is a key mechanism responsible for the formation of cell structures, have demonstrated this effect is relevant for h ~10–100 μm. As reported in studies [7,8,15,16], the thickness of molten layers ranges from ~10 to 80 μm in electron beam irradiation. This fact suggests thermoelectric effects represent one of the basic formation mechanisms of micro- and nanodimensional structure elements in electron beam processing.

On the other hand, a periodical arrangement of atoms of the base material and alloying elements in high-speed crystallization cells and columns (Figure 1a,d) might stem from the arising and developing Mullins–Sekerka instability [21–23], which is a result of the inhomogeneous temperature and concentration of an alloying element on the crystallization front. However, this mechanism fails to explain the importance of substance evaporation and thermoelectrical effects in electron-beam treatment. Further research should be done to study this issue.

Thus, micro- and nanodimensional structures may form in the process of electron beam irradiation owing to the emergent combination of thermal, concentration, evaporation and capillary, and thermoelectric instabilities. Within this context this work seeks to investigate laws and conditions supporting the formation of the above structures when processing by low energy high current millisecond electron beams.

## 2. Problem Formulation

Generally, conditions initiating the instability of small perturbations on the melt–plasma interface are specified via searching for nontrivial solutions to homogenous differential equations with homogenous kinematic and dynamic boundary conditions. The existence of a parameter range providing such solutions means a spontaneous transition of the substance into a new state is possible [24]. Similarly to studies [11,12,17], we focus on a viscous heat conducting and evaporating liquid, which occupies a layer $-h < z < 0$ and $-\infty < x < +\infty$ on a free surface $z = \eta(x, t)$. If affected by an electron beam, a temperature profile in the liquid layer is $T_0(z)$, where $T_0$—nonperturbed temperature (Figure 2). These profiles change over time due to thermal conductivity and diffusion, but if a characteristic time of evolving perturbations is shorter than their transformation period, a function $T_0(z)$ can be considered constant in the instability analysis [10,17]. A melt temperature is a sum of a nonperturbed $T_0(z)$ and perturbed $T(x,z,t)$ temperature. Assume a wavenumber vector of perturbed temperature and melt velocity is directed similarly to the X axis, these perturbations depend therefore on coordinates $x$ and time $t$ varying according to a function $\exp(\omega t - ikx)$, where $k$—wavenumber, $\omega$—complex cyclic frequency calculated as $\omega = \alpha + i\Omega$, $\alpha$—growth rate of perturbations, $\Omega$—cyclic frequency. For a temperature gradient, as mentioned in Introduction, a thermoelectric filed arises in a liquid

metal [18–20] and contributes to the convective melt flow, it is determined from $E = \gamma \nabla T$, where $\gamma$ —thermoelectric coefficient, $\nabla T$—temperature gradient.

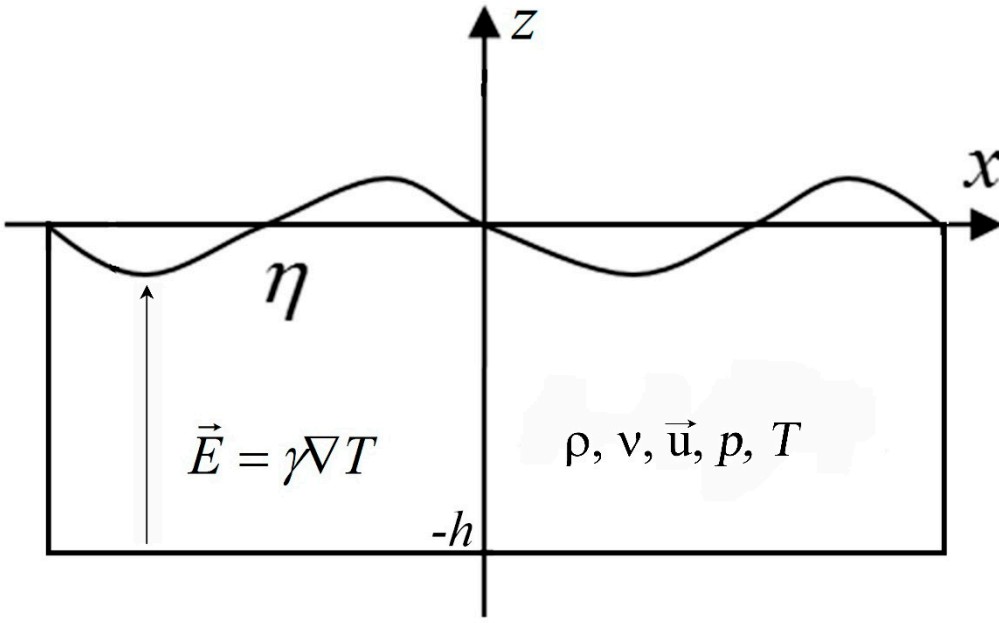

**Figure 2.** Scheme of combination instability formation.

A system of the Navier–Stokes equation and convective thermal conductivity equation is linearized and written as follows:

$$\frac{\partial u}{\partial t} = -\frac{1}{\rho}\frac{\partial p}{\partial t} + v\left(\frac{\partial^2 u}{\partial x^2} + \frac{\partial^2 u}{\partial z^2}\right),$$

$$\frac{\partial w}{\partial t} = -\frac{1}{\rho}\frac{\partial p}{\partial t} + v\left(\frac{\partial^2 w}{\partial x^2} + \frac{\partial^2 w}{\partial z^2}\right), \quad \frac{\partial u}{\partial x} + \frac{\partial w}{\partial z} = 0 \tag{1}$$

$$\frac{\partial T}{\partial t} + wG_0 = \chi\left(\frac{\partial^2 T}{\partial x^2} + \frac{\partial^2 T}{\partial z^2}\right)$$

where $u$, $w$ —disturbance vector components of velocity, $\rho$—density, $v$—kinematic viscosity, $\chi$—temperature conductivity, $G_0$—temperature gradient, $p$, $T$, $\eta$— pressure and temperature perturbations. No elements in the Equation (1) are related to the electric field, which is acceptable for conductive liquids [25]. Here, an electric field effect is included into boundary conditions on the surface $z = 0$:

$$\rho v\left(\frac{\partial u}{\partial z} + \frac{\partial w}{\partial x}\right) + \tau_{xz}^{el} = \frac{\partial \sigma}{\partial x},$$

$$-(p + p_e) + 2\rho v\frac{\partial w}{\partial z} + p'_v T = \sigma_0 \frac{\partial^2 \eta}{\partial x^2} \tag{2}$$

where $\eta$ —perturbation of the surface along the axis z, $\sigma = \sigma_0 + \sigma_T(T - T_m)$—surface tension, $\sigma_0$—surface tension at a melting temperature of the material $T_m$, $\sigma_T$—temperature coefficient of the surface tension, $p_e$—electric field–melt pressure, $\tau_{xz}^{el}$—tangential component of Maxwell stress tensor, $p'_v$—temperature coefficient of the evaporation pressure.

The kinematic boundary conditions and those for the thermal conductivity equation are formulated as follows:

$$z = 0: w = \frac{\partial \eta}{\partial t}, \quad \frac{\partial T}{\partial z} = 0.$$

$$z = -h: u = w = 0, \quad T = 0 \tag{3}$$

Electric elements of the boundary conditions (2) are determined on the following grounds [25,26]: electric field strength is $E_0$ on the stable horizontal surface of a liquid, the electric potential therefore $\varphi_0 = -E_0 z$. If a perturbation arises, the electric potential takes the form: $\varphi = \varphi_0 + \varphi_1$, where $\varphi_1$—small potential perturbation satisfying the Laplace equation $\Delta\varphi_1 = 0$ on the condition of $z \to +\infty$, $\varphi_1 \to 0$. This perturbation has a form of the normal mode with an amplitude damping on the axis $z$

$\phi_1 = C \exp(-kz) \exp(\omega t - ikx)$. Given that the electric potential along the wavelength is 0, we obtain $-E_0\eta + \phi_1(z = \eta) = 0$, i.e., $\phi_1(z = 0) = E_0\eta$, and the electric field pressure at $E_z = E_0 + k\phi_1$ takes the form: $p_e = \frac{\varepsilon\varepsilon_0}{2}\left(E_0^2 + 2kE_0^2\eta + k^2E_0^2\eta^2\right)$, where $\varepsilon$—dielectric permittivity, $\varepsilon_0$—electric constant. Omitting elements of the second order of smallness, $p_e = \varepsilon\varepsilon_0 kE_0^2\eta$ is obtained. A tangential component in the Maxwell stress tensor for $z = \eta$ is written $\tau_{xz}^{el} = \varepsilon\varepsilon_0 E_0 E_{1x}$, where $E_{1x} = -\frac{\partial \phi_1}{\partial x} = -E_0\frac{\partial \eta}{\partial x}$. In this case boundary conditions (2) are formulated:

$$z = 0: \ \rho v\left(\frac{\partial u}{\partial z} + \frac{\partial w}{\partial x}\right) = \sigma_T\frac{\partial T}{\partial x} + \varepsilon\varepsilon_0 E_0^2\frac{\partial \eta}{\partial x}$$

$$-(p + \varepsilon\varepsilon_0 kE_0^2\eta) + 2\rho v\frac{\partial w}{\partial z} + p'_v T = \sigma_0\frac{\partial^2 \eta}{\partial x^2} \tag{4}$$

Assume the surface displacement amplitude $\eta_0 << h$. So, it is possible to replace $z = -h$ with $z \rightarrow -\infty$. A solution to the Equation system (1) is sought in the form:

$$u(x,z,t) = -\frac{i}{k}(A_1 k \exp(kz) + A_2 k_1 \exp(k_1 z)) \exp(\omega t - ikx)$$

$$w(x,z,t) = (A_1 \exp(kz) + A_2 \exp(k_1 z)) \exp(\omega t - ikx)$$

$$p(x,z,t) = \frac{\rho v}{k}(k^2 - k_1^2)A_1 \exp(kz) \exp(\omega t - ikx) \tag{5}$$

$$T(x,z,t) = \frac{G_0}{\omega}((A_1 k - A_2 k_1 \delta)\exp(k_2 z) + A_2\delta\exp(k_1 z) - A_1\exp(kz))\exp(\omega t - ikx)$$

$$\eta = \eta_0 \exp(\omega t - ikx)$$

where $k_1 = \left(k^2 + \frac{\omega}{v}\right)^{1/2}$, $k_2 = \left(k^2 + \frac{\omega}{\chi}\right)^{1/2}$. Substituting (5) into boundary conditions (3) and (4), a system of linear algebraic equations in constants $A_1$ and $A_2$ is obtained:

$$\left(1 + \frac{k_1^2}{k^2} + \frac{1}{\omega}\left(\omega_p\left(\frac{k}{k_2} - 1\right) + \frac{\omega_c^2}{\omega_v}\right)\right)A_1 + \left(\frac{2k_1}{k} + \frac{1}{\omega}\left(\omega_p\delta\left(1 - \frac{k_1}{k_2}\right) + \frac{\omega_c^2}{\omega_v}\right)\right)A_2 = 0$$

$$\left(2 + \frac{\omega_T}{\omega}\left(1 + \frac{k}{k_2}\right) + \frac{\omega_E}{\omega}\right)A_1 + \left(1 + \frac{k_1^2}{k^2} + \frac{\omega_T\delta}{\omega}\left(\frac{k_1}{k_2} - 1\right) + \frac{\omega_E}{\omega}\right)A_2 = 0 \tag{6}$$

where $\omega_T = \frac{\sigma_T G_0}{\rho v}$, $\omega_v = vk^2$, $\omega_c^2 = \frac{\sigma_0 k^3}{\rho} - \frac{\varepsilon\varepsilon_0 E_0^2 k^2}{\rho}$, $\omega_p = \frac{p'_v G_0}{\rho vk}$, $\delta = \frac{\mathrm{Pr}}{1-\mathrm{Pr}}$, $\mathrm{Pr} = \frac{v}{\chi}$– Prandtl number. The temperature gradient is calculated as follows:

$$G_0 = \frac{q - q_{out}}{\kappa}, \tag{7}$$

where $q = \frac{E_s}{t_0}$ —power density of an electron beam, $E_s$—surface power density of an electron beam, $q_{out}$—evaporation-conditioned surface power density, $\kappa$—thermal conductivity of a liquid metal. The role of evaporation for the power density [27] is calculated as follows:

$$q_{out} = 10^6 \frac{A_{out}}{\sqrt{2\pi k_B T_m m}} \exp\left[\frac{A_{out}}{k_B}\left(\frac{1}{T_V} - \frac{1}{T_m}\right)\right] \tag{8}$$

where $A_{out}$ —work function of an atom escaping from the melt, $k_B$ —Boltzmann constant, $m$—atom mass, $T_V$—evaporation temperature.

The existence of a nontrivial solution to the system (6) needs its determinant to be zero, which in turn is a dispersion equation:

$$
\begin{aligned}
&R_T - R_E - R_V - R_\sigma = 0 \\
&R_\sigma = \omega^2\left((\omega + 2\omega_v)^2 + \omega_c^2\right) - \frac{4\omega^2\omega_v^2 k_1}{k}, \\
&R_T = \omega_T\omega_v\left(\delta\left(1 - \frac{k_1}{k_2}\right)\left(2\omega\omega_v + \omega^2 + \omega_c^2\right) + \left(1 - \frac{k}{k_2}\right)\left(\frac{2k_1\omega\omega_v}{k} + \omega_c^2\right)\right), \\
&R_E = -\omega_E\omega_v\omega\left(\left(1 - \frac{k_1}{k}\right)\omega_v + \omega\right), \\
&R_V = \omega_p\omega_v\begin{pmatrix}\left(\delta\left(1 - \frac{k_1}{k_2}\right) + \left(1 - \frac{k}{k_2}\right)\right)\omega_E\omega_v + 2\delta\omega\omega_v\left(1 - \frac{k_1}{k_2}\right) + \\ + (2\omega_v + \omega)\omega\left(1 - \frac{k}{k_2}\right)\end{pmatrix}
\end{aligned}
\tag{9}
$$

The Prandtl number for liquid metals takes a value $Pr << 1$. In this case the dispersion Equation (9) is formulated:

$$
\begin{aligned}
&\frac{\omega_T \mathrm{Pr}}{2}\left(2\omega\omega_v + \omega_c^2\right)\left(\omega + 2\omega_v\left(1 - \frac{k_1}{k}\right)\right) + \\
&+ \frac{\omega_p \mathrm{Pr}}{2}\left(\omega_E\omega_v\left(\omega + 2\omega_v\left(1 - \frac{k_1}{k}\right) + \omega\left(\omega^2 + 2\omega\omega_v + 4\omega_v^2\left(1 - \frac{k_1}{k}\right)\right)\right)\right) - \\
&- \omega_E\omega_v\omega\left(\left(1 - \frac{k_1}{k}\right)\omega_v + \omega\right) - \omega^2\left((\omega + 2\omega_v)^2 + \omega_c^2\right) + \frac{4\omega^2\omega_v^2 k_1}{k} = 0
\end{aligned}
\tag{10}
$$

Substituting $z = \frac{k_1}{k}$ and $\omega = \omega_v(z^2 - 1)$, an Equation (10) is written in the form:

$$
\begin{aligned}
&\left(C^2 + 2(z^2 - 1)\right)C_2 + \left(2C_4 + (z^2 + 2z + 3)(z^2 - 1)\right)C_3 - 2(z^2 + 1)C_4 - \\
&- (z + 1)^2\left(C^2 + (z^2 + 1)^2 - 4z\right) = 0
\end{aligned}
\tag{11}
$$

where $C = \frac{\omega_c}{\omega_v}$, $C_2 = \frac{\omega_T \mathrm{Pr}}{2\omega_v}$, $C_3 = \frac{\omega_p \mathrm{Pr}}{2\omega_v}$, $C_4 = \frac{\omega_E}{2\omega_v}$. Instable solutions comply with constraints $\mathrm{Re}(\omega) > 0$ and $\mathrm{Re}(z) > 0$. With the help of these solutions a relationship between the growth rate of perturbations $\alpha = \mathrm{Re}(\omega)$ and wavenumber (wavelength) may be established. Using this correlation, we determine a wavelength, when the instability ($\alpha = 0$) arises, and a value $\lambda$ for $\alpha = \alpha_m$. The instability of the plasma–melt interface is explored on the example of titanium and aluminum alloys. The data for calculations are provided in Table 1.

Table 1. Thermal physical properties of systems Ti-Y and Al-Si-Y [28–30].

| Symbol/Dimension | Ti-Y | Al-Si-Y | Property |
|---|---|---|---|
| $T_m$, K | 1628 | 850 | Melting temperature |
| $T_V$, K | 1810 | 1270 | Evaporation temperature |
| $\rho_L$, kg/m$^3$ | 4120 | 2398 | Liquid phase density |
| $\nu$, $10^{-7}$ m$^2$/s | 9.0 | 3.5 | Viscosity |
| $\chi$, $10^{-5}$ m$^2$/s | 0.91 | 3.3 | Thermal diffusivity |
| $D$, $10^{-7}$ m$^2$/s | 8.9 | 3.4 | Diffusion coefficient |
| $\sigma$, N/m | 1.64 | 0.87 | Surface tension |
| $\sigma_T$, $10^{-3}$ N/(m·K) | −0.238 | −0.35 | Temperature coefficient of surface tension |
| $\sigma_c$, $10^{-2}$ N/(m·m$^{-2}$) | −1.0 | 2.31 | Concentration coefficient of surface tension |
| $\kappa$, W/(m·K) | 33.1 | 86 | Thermal conductivity |
| $A$, eV | 4.0 | 3.23 | Work function |
| $m$, $10^{-26}$ kg | 7.97 | 4.49 | Atom mass |

The thermoelectric coefficient for a temperature above the Debye one is estimated by the formula $\gamma = f\frac{k_B T_D}{eT}$ [31], where $T_D$—Debye temperature, $e$—charge carrier, $f$—constant with values ensuring $\gamma \sim 100$ µV/K.

The correlation between the growth rate of perturbations on the interface and wavelength is estimated for an electron beam power density of 35 J/cm$^2$ and a pulse time of 150 µs.

## 3. Results and Discussion

First, we consider a situation, when thermoelectric effects and evaporation pressure may be omitted. Here, a gradient of nonperturbed temperature $G_0 = 3.66 \cdot 10^7$ K/m—start of the instability—occurs in titanium alloys when the wavelength is 38 µm, and a maximal growth rate is registered for a wavelength of 113 µm (Figure 3 curve 1). A value of the thermoelectric field for $\gamma \sim 100$ µV/K ranges up to $E_0 \sim 10^3$ V/m, in case of such a field value the thermoelectric effect is almost irrelevant to a growth rate of perturbations on the interface. The reason is that the thermoelectric coefficient is calculated for a solid in the pre-melting state. In a liquid substance charge, carrying is highly dependent on the convective flow, which intensifies the thermoelectric effect. Therefore, the value $\gamma$ is ~$10^4$–$10^5$ µV/K; and electric field strength $E_0 \sim 10^5$–$10^6$ V/m, respectively. For such field values the growth rate of perturbations increases significantly (Figure 3 curve 2), whereas the critical wavelength is two times shorter (17 µm). A maximal growth rate is registered for $\lambda = 50$ µm. For an evaporation pressure of $2 \cdot 10^5$ Pa the critical wavelength increases to 22 µm with a maximal growth rate for a wavelength of 92 µm (Figure 3 curve 3). In this case the thermoelectric effect makes no contribution to instability on the interface (Figure 3 curve 4). In aluminum alloys the same trend is observed (Figure 3). Ignoring evaporation pressure and thermoelectric effects for a nonperturbed temperature gradient $G_0 = 1.81 \times 10^7$ K/m, the critical wavelength is 39 µm, and a maximal growth rate of perturbations is attributable to a wavelength of 155 µm (Figure 4 curve 1). With thermoelectric effects, we obtain $\lambda_{cr} = 13$ µm and $\lambda_m = 53$ µm (Figure 4 curve 2). For an evaporation pressure of $2 \times 10^5$ Pa—$\lambda_{cr} = 17$ µm and $\lambda_m = 87$ µm. The thermoelectric effect shifts the maximal growth rate to the wavelength $\lambda_m = 76$ µm, and the critical wavelength is $\lambda_m = 18$ µm in these conditions (Figure 4 curve 3 and 4). From the experimental data it can be seen [7,8] that cell structures develop due to the electron beam irradiation varying in a range from 100 to 600 nm, depending on processing conditions. The low frequency approximation (10) has shown this range of wavelengths is possible for electric field strength $E_0 \sim 10^6$–$10^7$ V/m (Figure 5 curve 1 and 2). In titanium alloys $\lambda_{cr} = 0.17$ µm and $\lambda_m = 0.3$ µm, and in aluminum alloys—$\lambda_{cr} = 0.13$ µm and $\lambda_m = 0.26$ µm.

In the calculations presented the effect of a doping element on the melt flow is ignored. In fact, the occurrence of doping elements is essential to the melt surface tension and to the thermocapillary instability as well [32]. Depending on the nature of a base material, a doping element may be a surface-active substance, which reduces (surface-inactive substance) or increases the surface tension of the melt or have no effect on the surface tension of a material. This study investigates the influence of an yttrium concentration gradient on the thermocapillary flow of a titanium or aluminum melt. From the experimental data on the surface tension it is apparent [33] that yttrium is a surface-active substance as related to titanium, whereas it is a surface-inactive substance to aluminum. Including the influence of a doping element concentration, the surface tension vs. temperature and concentration relationship is formulated as follows:

$$\sigma = \sigma_0 + \sigma_T(T - T_m) + \sigma_c(C - C_0), \tag{12}$$

where $\sigma_C$—concentration coefficient of the surface tension, $C$—surface concentration. A convective diffusion equation needs to be added to the system (1):

$$\frac{\partial C}{\partial t} + wG_1 = D\left(\frac{\partial^2 C}{\partial x^2} + \frac{\partial^2 C}{\partial z^2}\right), \tag{13}$$

where $D$—diffusion coefficient, $G_1$—concentration gradient. The boundary conditions for tangential stresses are written:

$$\rho v\left(\frac{\partial u}{\partial z} + \frac{\partial w}{\partial x}\right) = \sigma_T \frac{\partial T}{\partial x} + \sigma_C \frac{\partial C}{\partial x} + \varepsilon\varepsilon_0 E_0^2 \frac{\partial \eta}{\partial x} \tag{14}$$

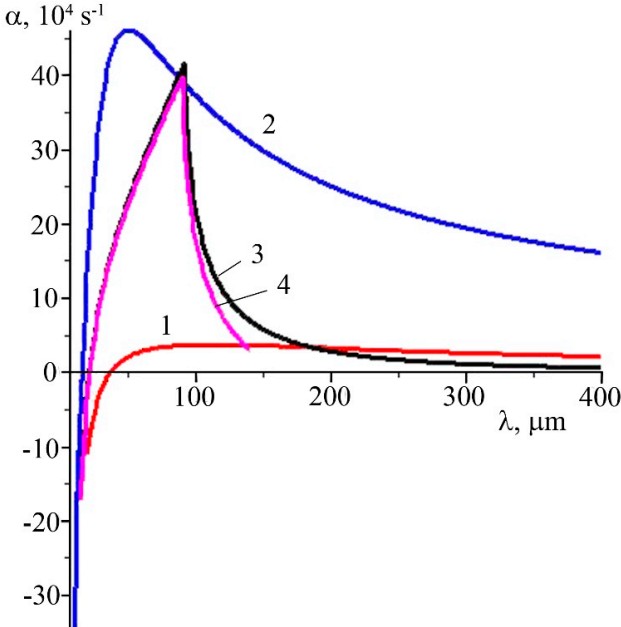

**Figure 3.** A growth rate characteristic of perturbations on the plasma–titanium melt interface found when solving the Equation (11): (**1**) ignoring thermoelectric and evaporation and capillary effects, (**2**) including the thermoelectric effect only, (**3**) with regard to thermal and evaporation and capillary instabilities, (**4**) involving thermal, evaporation and capillarity instabilities and the thermoelectric effect.

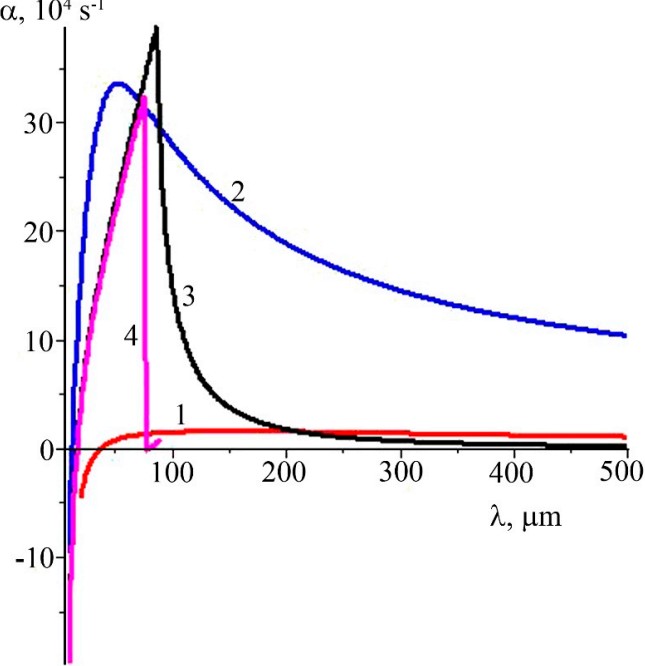

**Figure 4.** A growth rate characteristic of perturbations on the plasma–silumin melt interface found when solving the Equation (11): (**1**) ignoring thermoelectric and evaporation and capillary effects, (**2**) including the thermoelectric effect only, (**3**) with regard to thermal and evaporation and capillary instabilities, (**4**) involving thermal, evaporation and capillarity instabilities and the thermoelectric effect.

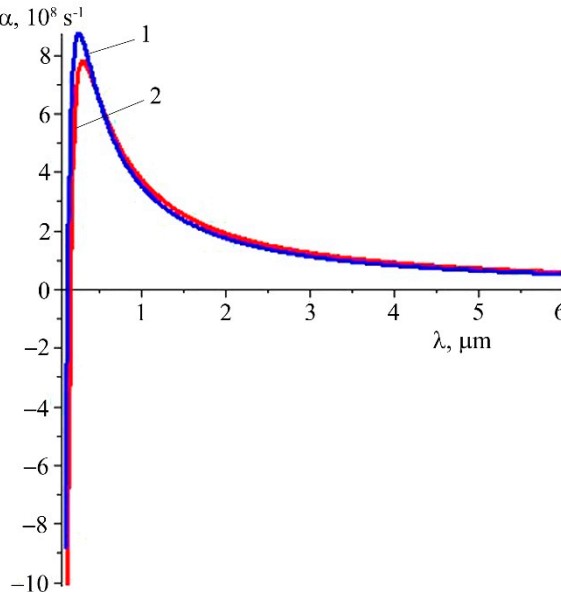

**Figure 5.** A growth rate characteristic of perturbations on the plasma–melt interface for a thermoelectric constant of $10^{-1}$ V/K: (**1**) Ti-Y system; (**2**) Al-Si-Y system.

To consider the effect of a doping element concentration in the low frequency approximation, a product $\omega_T Pr$ is to be replaced by $\omega_T Pr + \omega_C Sc$ in the dispersion Equation (10), where $Sc = \nu/D$ —the Schmidt number, $\omega_C = \frac{\sigma_C G_1}{\rho \nu}$. The use of the low frequency approximation is reasonable provided that $Pr << 1$ and $Sc << 1$. For liquid metals the Schmidt number is ~10 [34]. It was assessed under the assumption of a stable liquid. In practice, the convective flow intensifies mass transfer processes; thus, the diffusion coefficient is accepted to be one order of magnitude higher than the kinematic viscosity coefficient. Figure 5 gives curves of a growth rate on the melt interface of yttrium-doped titanium for a surface concentration gradient of $G_1 = 10^6$ m$^{-2}$/m.

As can be seen, if thermoelectric and evaporation effects are ignored, the critical wavelength is 26 μm, and a maximal growth rate aligns with a wavelength of 74 μm (Figure 6, curve 1). The consideration of thermoelectric effects results in a critical wavelength of 17 μm with a maximal growth rate conforming to a wavelength of 48 μm (Figure 6, curve 2). For an evaporation pressure of $2 \times 10^5$ Pa $\lambda_{cr} = 11$ μm and $\lambda_m = 36$ μm thermoelectric effects cause a slightly increasing maximal growth rate of perturbations (Figure 6, curves 3 and 4). In aluminum alloys another phenomenon is observed for a concentration gradient of $G_1 = 10^6$ m$^{-2}$/m. If $\omega_E = 0$ and $\omega_p = 0$, the critical wavelength is 7 μm, and the maximal growth rate corresponds to 25 μm (Figure 7, curve 1). For $\omega_E \neq 0$ and $\omega_p = 0$ the critical wavelength is 7 μm, and the growth rate maximum is detected for wavelengths of 10 and 44 μm (Figure 7, curve 2). For $\omega_E = 0$ and $\omega_p \neq 0$ the critical wavelength is 7 μm, and a maximal growth rate is attained for a wavelength of 27 μm (Figure 7, curve 3). For $\omega_E \neq 0$ and $\omega_p \neq 0$ the growth rate function also displays two maximums (Figure 7, curve 4). The first maximum corresponds with a wavelength of 10 μm and the second one—110 μm. The occurrence of the first maximum might result from the interaction of concentration and capillary and thermoelectric instabilities, the second maximum might be caused by a temperature gradient. The results obtained explain adequately the formation of surface-periodic structures in a micrometer range. Within the low frequency approximation, similarly to the above considered situation, submicro- and nanodimensional cell structures may develop for a thermoelectric coefficient ~$10^{-1}$ V/K or an evaporation pressure ~$10^{11}$ Pa. Thus, the inclusion of the yttrium concentration makes a spectrum of capillary waves in the Al-Si-Y system more complex, whereas this effect is not detected in the Ti-Y system.

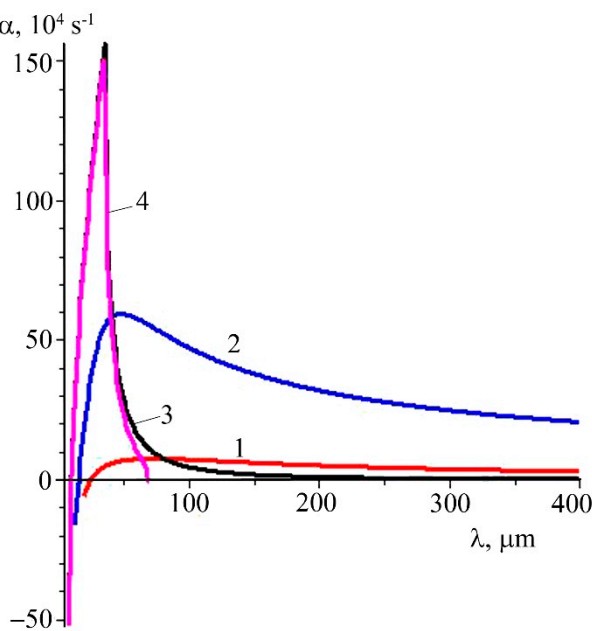

**Figure 6.** A growth rate characteristic of perturbations on the plasma–titanium melt interface for a gradient of surface concentration of 10 m$^{-2}$/m found when solving the Equation (11): (**1**) ignoring thermoelectric and evaporation and capillary effects, (**2**) including the thermoelectric effect only, (**3**) with regard to thermal and evaporation and capillary instabilities, (**4**) involving thermal, evaporation and capillarity instabilities and the thermoelectric effect.

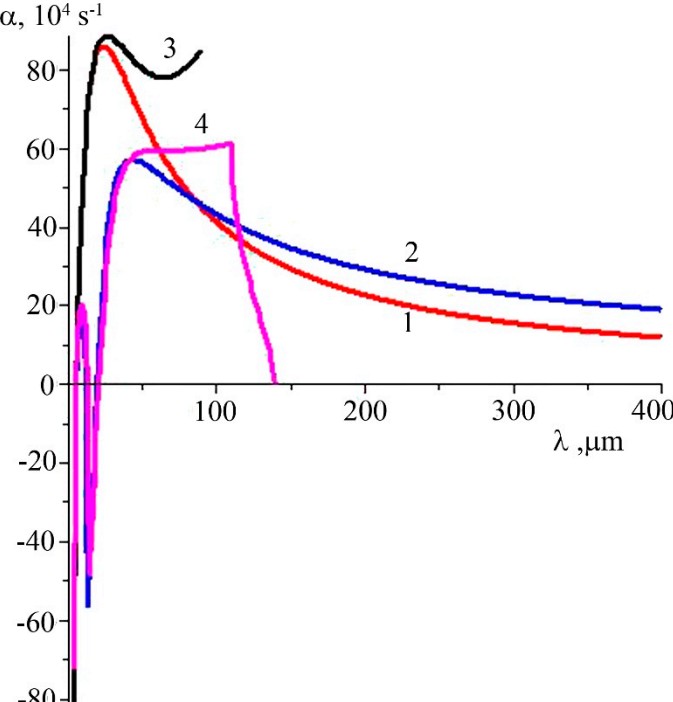

**Figure 7.** A growth rate characteristic of perturbations on the plasma–silumin melt interface for a gradient of surface concentration of 10 m$^{-2}$/m found when solving the Equation (11): (**1**) ignoring thermoelectric and evaporation and capillary effects, (**2**) including the thermoelectric effect only, (**3**) with regard to thermal and evaporation and capillary instabilities, (**4**) involving thermal, evaporation and capillarity instabilities and the thermoelectric effect.

An analysis of the dispersion Equation (10) has revealed that the low frequency approximation even with included thermoelectric and evaporation capillary effects cannot provide an adequate explanation why a micro-and nanodimensional cell crystallization structure forms; we seek therefore for functions of the perturbations growth rate on the base of solutions to the dispersion Equation (9). For this purpose, we use the substitution $k_1 = z_1 k$, $k_2 = z_2 k$ and $\omega = \omega_v(z^2 - 1)$, $z_2 = \left(1 + \Pr(z_1^2 - 1)\right)^{1/2}$. This substitution transforms the Equation (9) into an algebraic equation of the 16th degree in $z_1$, which is too lengthy to be given here. Roots of this equation satisfying the constraints $Re(\omega) > 0$, $Re(z_1) > 0$, $Re(z_2) > 0$ are instable. Figure 8 provides the data of how the growth rate of perturbations depends on a wavelength in the Ti-Y system.

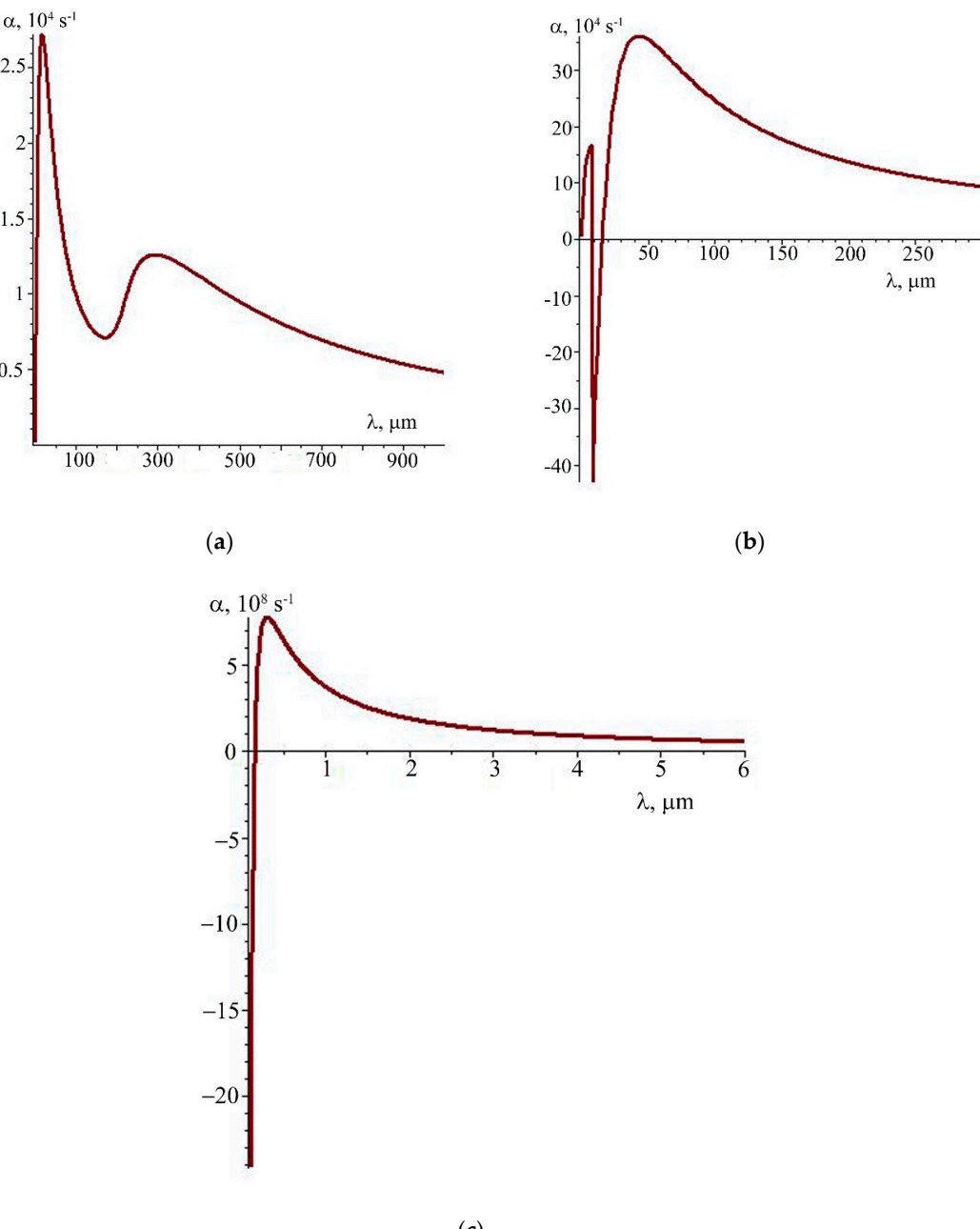

(a)

(b)

(c)

**Figure 8.** A growth rate characteristic of perturbations on the plasma–melt interface for the Ti-Y system found when solving the Equation (9): (**a**) ignoring thermoelectric and evaporation and capillary effects, (**b**) for thermoelectric field strength of $10^5$ V/m, (**c**) for thermoelectric field strength of $10^6$ V/m.

　　　It is apparent that without thermoelectric and evaporation capillary effects there are two maximums on this curve. The first maximum is found for a wavelength of 18 μm, and the second one appears if the wavelength is 294 μm (Figure 8a). If a thermoelectric field ~$10^5$ V/m arises, the first maximum is detected for a wavelength of 8 μm, and the second one 43 μm (Figure 8b). An increase in thermoelectric field strength up to $10^6$ V/m eliminates the second maximum, and makes the first one conform to a wavelength of 0.31 μm (Figure 8c). In the Al-Si-Y system we find similar transformations in the growth rate vs. wavelength relationship (Figure 9).

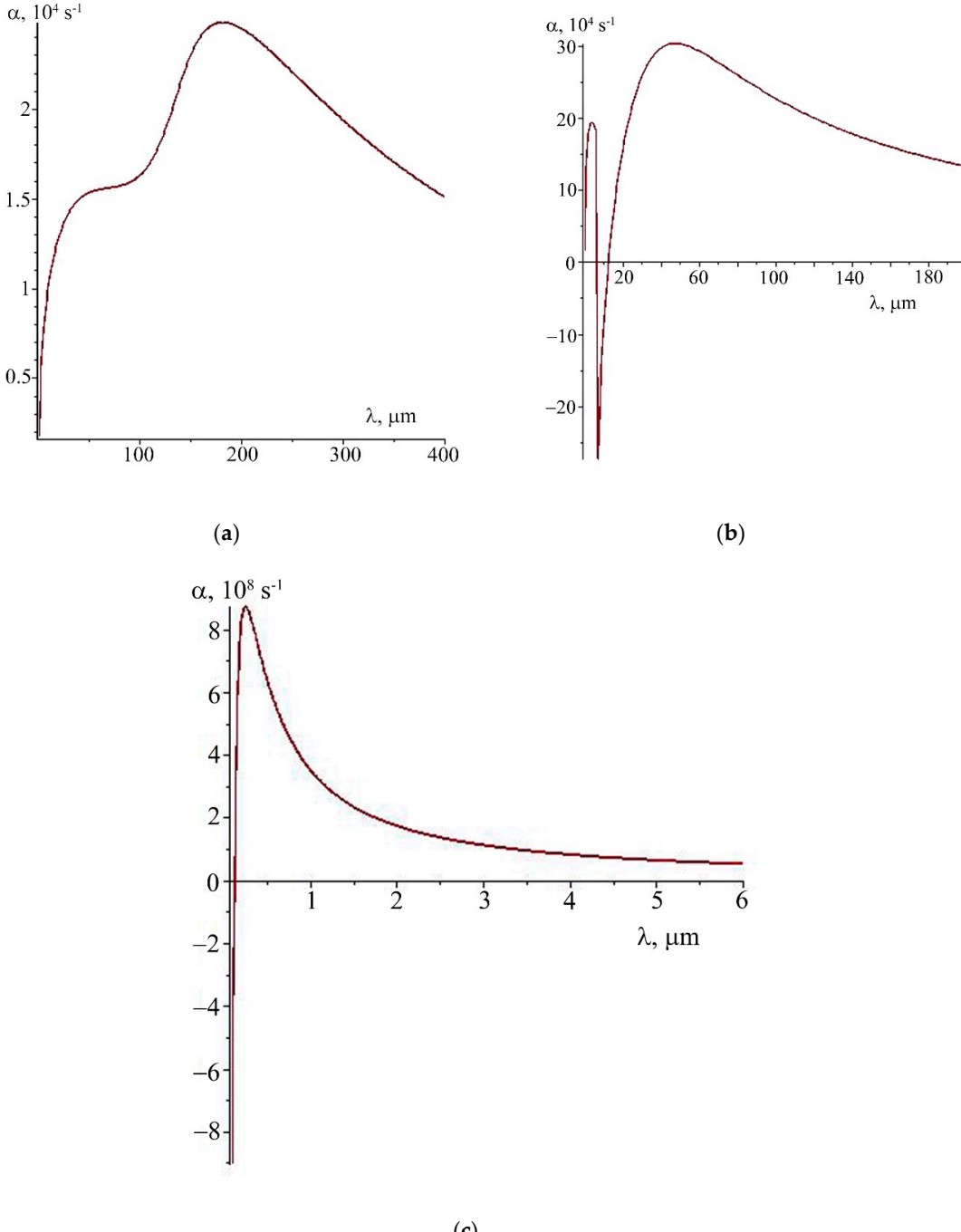

(**a**)　　　　　　　　　　　　　　　　　　　　　　　(**b**)

(**c**)

**Figure 9.** A growth rate characteristic of perturbations on the plasma–melt interface for the Al-Si-Y system found when solving the Equation (9): (**a**) ignoring thermoelectric and evaporation and capillary effects, (**b**) for thermoelectric field strength of $10^5$ V/m, (**c**) for thermoelectric field strength of $10^6$ V/m.

Without thermoelectric effects the growth rate maximum is for a wavelength of 182 μm (Figure 9a), and two maximums are detected for wavelengths of 4 μm and 48 μm, given that field strength is $10^5$ V/m (Figure 9b). For $E_0 \sim 10^6$ V/m—$\lambda_m = 0.26$ μm (Figure 9c). In the Ti-Y system the maximum is shifted to a wavelength of 9.6 μm for an evaporation pressure of $2 \times 10^5$ Pa, ignoring thermoelectric effects. Given that $\omega_E \neq 0$ and $\omega_p \neq 0$, a maximal growth rate is registered for wavelengths of 4.2 μm and 19.6 μm. In the Al-Si-Y system similar transformations are observed.

## 4. Conclusions

The detected correlations between the thermoelectric effect and early instability stage of the thermocapillary melt flow in the electron beam irradiation suggest that surface micro- and nanodimensional structures form due to the combination of thermal, concentration, evaporation and capillary and thermoelectric instabilities. The findings provide some support for the significant importance of thermoelectric and concentration and capillary phenomena to the growth rate of perturbations on the plasma–melt interface. The low frequency approximation in the Al-Si-Y system (9) demonstrates two maximums of this correlation, whereas only one maximum is detected in the Ti-Y system. A growth rate maximum is attributed to submicro- and nanometer wavelengths for a thermoelectric coefficient $\sim 10^{-1}$ V/K both in Ti-Y and Al-Si-Y system. The results obtained may be useful when searching for appropriate conditions of the electron beam irradiation to further the formation of micro- and nanostructures.

**Author Contributions:** S.N., model building and performance. V.S., designed the model and analyzed the simulation results. S.K., the formulation of the problem of power influence on titanium and aluminum alloys, A.G., the calculations of thermocapillary and thermoelectric models. V.G., main comments on the revision of model and analysis. All authors have read and agreed to the published version of the manuscript.

**Funding:** This work was supported by the President grant for State Support to young researches [grant number MK-118.2019.2].

**Conflicts of Interest:** The authors declare no conflict of interest.

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
