# Peer review of "Formation Mechanism of Micro- and Nanocrystalline Surface Layers in Titanium and Aluminum Alloys in Electron Beam Irradiation"

_metals, doi:10.3390/met10101399_

Round 1

Reviewer 1 Report

The manuscript proposes a theoretical model to explain the characteristic size of microstructural features seen in the surface layers of electron beam irradiated Ti-Y and Al-Si-Y. Predictive models describing the formation of microstructural features as a function of experimental/processing conditions are useful because microstructure and mechanical properties are strongly correlated. The ability to predict the microstructure is a step closer to predicting the mechanical properties which are of high importance for the applications of these materials.

The calculations are presented in the manuscript are remarkable in their detail and comprehensiveness. 

On the other hand, there is hardly any experimental data shown to which all the modeling results can be compared. The images in Fig 1 are not sufficient. Fig 1a shows a structure where one can see a layer having a periodic structure, but Fig 1b simply looks like a typical grain structure. The quality of the manuscript would better if more TEM images would be presented showing the microstructures in question.

Also, it would make the manuscript stronger if measurements would be conducted on the TEM images and one would have statistics available on the microstructural features in questions. The authors refer to some numbers when talking about microstructural features, but reporting averages and standard deviations (or something similar) would be better, coupled with showing more TEM images which were used to obtain these values.

The lack of convincing experimental data, which would validate the existence of the model in the first place, is the main weakness of the manuscript.

Other suggestions for improvement:

  • It would be beneficial to add plots and/or diagrams to the Problem Formulation section to explain the model through them, as it would significantly enhance readability.
  • Line 260: The authors should provide that very long equation as supplementary material.
  • The label of the X axis on Fig 5 is missing.

Author Response

Dear Editor and Reviewers,

Thank you very much for your reply and help. We truly appreciate all the constructive comments and suggestions from all reviewers. We have adopted all the suggestions in our revised manuscript entitled “Formation mechanism of micro-and nanocrystalline surface layers in titanium and aluminum alloys in electron beam irradiation”. The following are our point-to-point responses to the reviewers’ comments:

On the other hand, there is hardly any experimental data shown to which all the modeling results can be compared. The images in Fig 1 are not sufficient. Fig 1a shows a structure where one can see a layer having a periodic structure, but Fig 1b simply looks like a typical grain structure. The quality of the manuscript would better if more TEM images would be presented showing the microstructures in question.

Also, it would make the manuscript stronger if measurements would be conducted on the TEM images and one would have statistics available on the microstructural features in questions. The authors refer to some numbers when talking about microstructural features, but reporting averages and standard deviations (or something similar) would be better, coupled with showing more TEM images which were used to obtain these values.

The lack of convincing experimental data, which would validate the existence of the model in the first place, is the main weakness of the manuscript.

We agree with the point to validate the results obtained in the study more electron-microscopic data are necessary and statistics on dimensions of structure elements, e.g. the average and standard deviation are to be presented. Electron-microscopic data shown in Figure 1 were taken from papers [Sosnin, K.V.; Ivanov, Y.F.; Glezer A.M.; Gromov, V.E.; Raykov S.V.; Budovskikh, E.A. Combined Electron–Ion Plasma Doping of a Titanium Surface with Yttrium: Analyzing Structure and Properties. Bulletin of the Russian Academy of Sciences. Physics. 2014, 78, 1183 ‒ 1187. https://doi.org/ 10.3103/S1062873814110264] and [Zaguliaev, D.; Konovalov, S.; Ivanov, Y.; Gromov, V. Effect of electron-plasma alloying on structure and mechanical properties of Al-Si alloy. Appl. Surf. Sci. 2019, 498, 143767. https://doi.org/10.1016/j.apsusc.2019.143767 ], which reported on the range of cell dimensions. A validation criterion selected for this model was whether a wavelength with a maximal growth rate of perturbations is within this range. Unfortunately, the authors haven’t presented any bar graphs for the size distribution of grains where we might find data to determine an average grain size and its most probable value. Nevertheless, the proposed mathematical model allows identifying a wavelength with a maximal growth rate of perturbations; in addition, the most probable size of structure elements is needed for the reliable comparison with simulation results. The study [ Sarychev, V., Nevskii, S., Konovalov, S., Granovskii, A., Gromov, V. Thermocapillary model of formation of nanostructures on the surface irradiated by low-energy high-current electron beams (2019) Materials Research Express, 6 (7), paper No 076551 DOI: 10.1088/2053-1591/ab1526] and other works, e.g. [15, 16] have shown bar graphs for the cell size distribution in aluminum alloys irradiated with electron beams without pretreatment by electrical explosion plasma of yttrium powder; according to these data the sizes of grains vary in submicro- and nanocrystalline ranges. However, we doubt the correctness of comparing these data with simulation results.

In light of the Reviewer’s comments we added more electron-microscopic images. Figure 1 was supplemented with panoramic electron-microscopic data presenting a thickness of the silumin modified layer, as well as images evidencing this layer to consist of submicro-and nanocrystalline cells.   

Other suggestions for improvement:

It would be beneficial to add plots and/or diagrams to the Problem Formulation section to explain the model through them, as it would significantly enhance readability.

We thank the Reviewer for this comment the Problem Formulation Section should be supplemented with a diagram explaining an initial phase of the thermocapillary instability. This diagram was added. It should be noted a similar diagram was presented in our paper [12].

Line 260: The authors should provide that very long equation as supplementary material.

Unfortunately, we can’t provide this equation in the article for its awkwardness (see the Maple screenshot)

The label of the X axis on Fig 5 is missing.

We agree with this point. The label was added to the axis Х – λ, μm.

All corrections in the article are marked by color.

Kind regards,   Sergey Konovalov

Reviewer 2 Report

Lines 32-34: "The wear and fracture of machine parts begin on their surface because it is more supportive of developing crystal lattice defects than the material beneath it". This statement could be rephrased to have more clarity.

Table 1: Add references to the thermophysical properties listed.

Is there element segregation between cell cores and cell boundaries in cells shown in Figure 1?
If so, the cells could have formed also via Mullins-Sekerka instability

The manuscript should at least mention that it is possible that the solidification cells could rise due to Mullins-Sekerka instability, or present convicing arguments why Mullin-Sekerka instability is not a viable explanation. This is a rather well established mechanism of cell formation in e.g. laser welding and additive manufacturing technologies such as selective laser melting.
Briefly, the solute diffusion length is typically much smaller than the length scales associated with flow effects, and therefore solute diffusion and Mullins-Sekerka instability likely dominates this solidification technique. It is still possible that the behavior is different for these electron beam irradiation techniques.
It should be noted that while the laser melting creates a melt pool with strong currents and various modes of flow instabilities, there is good evidence that these flow effects can be ignored in analys of the solid-liquid interface morphology. These flow effects crucially affect the longer range temperature field, which then in turn influences the solidification conditions more indirectly.

Author Response

Dear Editor and Reviewers,

Thank you very much for your reply and help. We truly appreciate all the constructive comments and suggestions from all reviewers. We have adopted all the suggestions in our revised manuscript entitled “Formation mechanism of micro-and nanocrystalline surface layers in titanium and aluminum alloys in electron beam irradiation”. The following are our point-to-point responses to the reviewers’ comments:

Lines 32-34: "The wear and fracture of machine parts begin on their surface because it is more supportive of developing crystal lattice defects than the material beneath it". This statement could be rephrased to have more clarity.

We agree with this point raised by the Reviewer. The statement in Lines 32-34 has been revised: The material fracture is known to arise in the surface and near-surface layers because long-ranging stresses in them exceed the material strength limit.

Table 1: Add references to the thermophysical properties listed.

We thank the Reviewer for highlighting this point. We have added literature sources disclosing the thermophysical constants of the material to the list of references.

Is there element segregation between cell cores and cell boundaries in cells shown in Figure 1? If so, the cells could have formed also via Mullins-Sekerka instability

The manuscript should at least mention that it is possible that the solidification cells could rise due to Mullins-Sekerka instability, or present convincing arguments why Mullin-Sekerka instability is not a viable explanation. This is a rather well established mechanism of cell formation in e.g. laser welding and additive manufacturing technologies such as selective laser melting.

Briefly, the solute diffusion length is typically much smaller than the length scales associated with flow effects, and therefore solute diffusion and Mullins-Sekerka instability likely dominates this solidification technique. It is still possible that the behavior is different for these electron beam irradiation techniques.

It should be noted that while the laser melting creates a melt pool with strong currents and various modes of flow instabilities, there is good evidence that these flow effects can be ignored in analys of the solid-liquid interface morphology. These flow effects crucially affect the longer range temperature field, which then in turn influences the solidification conditions more indirectly.

The research outcomes reported in [7] demonstrate that a columnar layer (Figure 1a) has a stratified structure formed by titanium and yttrium containing columns; this observation indicates the phase segregation of the material. The study [8] has presented some evidence aluminum atoms are a prime component of high-speed crystallization cells. Atoms of silicon and yttrium are detected in thin layers. As the Reviewer correctly pointed out, cell could rise due to Mullins-Sekerka instability. We added some information on this mechanism into Introduction. Micro-and nanocrystalline cell structures appear to develop owing to the combination of thermocapillary, concentration-capillary, evaporation-capillary and Mullins-Sekerka instabilities. However, further research should be done to investigate this instability in micro- and nano-range with thermoelectrical effects and evaporation.

All corrections in the article are marked by color.

Kind regards, Sergey Konovalov

Reviewer 3 Report

The authors studied the formation of micro- and nanocrystalline layers in titanium and aluminum alloys in electron beam irradiation. The manuscript should be of interest to some readers and it is suitable for the publication in Metals journal. However, to be accepted for publication the following comments are required to be addressed:

Please Add link and change line 52.

Have you compared the simulation results with the experiment?

Author Response

Dear Editor and Reviewers,

Thank you very much for your reply and help. We truly appreciate all the constructive comments and suggestions from all reviewers. We have adopted all the suggestions in our revised manuscript entitled “Formation mechanism of micro-and nanocrystalline surface layers in titanium and aluminum alloys in electron beam irradiation”. The following are our point-to-point responses to the reviewers’ comments:

Please Add link and change line 52.

We agree with this point raised by the Reviewer. A reference to the literature source presenting the cell dimensions has been added.

Have you compared the simulation results with the experiment?

The theoretical data have been compared with the experimental findings. In this way we adjusted the model for both the concentration-capillary and evaporation- capillary effects were analyzed. Furthermore, a thermo-electrical coefficient was corrected.

All corrections in the article are marked by color.Kind regards, Sergey Konovalov

Round 2

Reviewer 1 Report

The Authors have addressed the most of the Reviewer's, except the one related to the Problem Formulation Section. They say they added a diagram to that section, but I cannot see it in the new version of the manuscript. A similar diagram shown in ref [12] (Fig 3 in Mater. Res. Express 6 (2019) 026540) would be beneficial to this manuscript too, in the Reviewer's opinion.

Author Response

Dear Editor and Reviewers,

Thank you very much for your reply and help. We truly appreciate all the constructive comments and suggestions from all reviewers. We have adopted all the suggestions in our revised manuscript entitled “Formation mechanism of micro-and nanocrystalline surface layers in titanium and aluminum alloys in electron beam irradiation”. The following are our point-to-point responses to the reviewers’ comments:

The Authors have addressed the most of the Reviewer's, except the one related to the Problem Formulation Section. They say they added a diagram to that section, but I cannot see it in the new version of the manuscript. A similar diagram shown in ref [12] (Fig 3 in Mater. Res. Express 6 (2019) 026540) would be beneficial to this manuscript too, in the Reviewer's opinion.

We agree with this comment of the reviewer. The scheme of instability origination have been added to the article.

All corrections in the article are marked by color.

Kind regards, Sergey Konovalov
